# Evaluating Bias and Fairness in Gender-Neutral Pretrained Vision-and-Language Models

**Laura Cabello    Emanuele Bugliarello    Stephanie Brandl    Desmond Elliott**
Department of Computer Science, University of Copenhagen, Denmark
{lcp,emanuele,brandl,de}@di.ku.dk

## Abstract

Pretrained machine learning models are known to perpetuate and even amplify existing biases in data, which can result in unfair outcomes that ultimately impact user experience. Therefore, it is crucial to understand the mechanisms behind those prejudicial biases to ensure that model performance does not result in discriminatory behaviour toward certain groups or populations. In this work, we define gender bias as our case study. We quantify bias amplification in pretraining and after fine-tuning on three families of vision-and-language models. We investigate the connection, if any, between the two learning stages, and evaluate how bias amplification reflects on model performance. Overall, we find that bias amplification in pretraining and after fine-tuning are independent. We then examine the effect of continued pretraining on gender-neutral data, finding that this reduces group disparities, *i.e.*, promotes fairness, on VQAv2 and retrieval tasks without significantly compromising task performance.

## 1 Introduction

As shown by Mitchell (1980) and Montañez et al. (2019), inductive biases are essential for learning algorithms to outperform random guessing. These task-specific biases allow algorithms to generalize beyond training data but, necessarily, they should not be conflated with prejudicial or unwanted biases. Unwanted bias, such as bias against demographic groups, can be found in many applications, from computer vision systems to natural language processing (NLP). Vision-and-language (V&L) models lie at the intersection of these areas, where one of the key challenges is deploying robust models to perform high-level reasoning based on the multimodal context instead of exploiting biases in data (Zhao et al., 2017).

Multiple studies (Lee et al., 2021; Hirota et al., 2022b; Zhou et al., 2022) have shown that V&L models leverage co-occurrences between objects

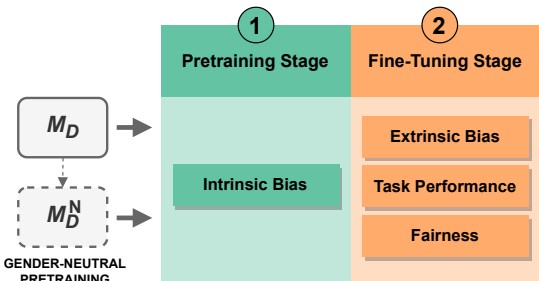

Figure 1: A V&L model pretrained on data $D$ ($M_D$) is further pretrained on gender-neutral multimodal data $D^N$, resulting in a gender neutral V&L model ($M_D^N$). Both models can then be used in a two-phase analysis: 1) bias amplification is measured on the intrinsic bias of pretrained models, and 2) bias amplification, task performance and fairness are evaluated on the extrinsic performance of fine-tuned models.

and their context to make predictions, and thus are susceptible to unwanted biases. However, these authors do not explore the broad landscape of V&L models and focus on biases in common visual datasets (Wang et al., 2022; Hirota et al., 2022a), only on pretrained models (Zhou et al., 2022) or only focus on one application, *e.g.*, image captioning (Hendricks et al., 2018; Hirota et al., 2022b) or semantic segmentation (Lee et al., 2021).

In this work, we investigate to what extent the unwanted bias in a V&L model is caused by the *pretraining data*. To answer this question, we focus on one important aspect of bias encoded in V&L models, namely *bias amplification*. Bias amplification occurs when a model exacerbates unwanted biases from the training data and, unlike other forms of bias, it is not solely attributed to the data, yet it can vary greatly during training (Hall et al., 2022).

We explore bias amplification in two encoder-only V&L models: LXMERT (Tan and Bansal, 2019) and ALBEF (Li et al., 2021), and the encoder-decoder model BLIP (Li et al., 2022). Specifically, we quantitatively and qualitatively

analyse the relationship between the bias encoded in pretrained models, and after fine-tuning on downstream tasks including visual question answering, visual reasoning and image–text retrieval.

While bias can be studied with respect to any protected attribute, the majority of NLP research has focused on (binary) gender (Sun et al., 2019; Stanczak and Augenstein, 2021; Shrestha and Das, 2022). We also use gender bias as our case study but different to previous work, we advocate for the inclusion of gender-neutral terms (Dev et al., 2021) and consider three gender categories based on visual appearance: male, female and gender-neutral (*e.g.*, PERSON). The use of both visual and grammatical gender information across V&L tasks is needed for identifying the target of, for example, a question. But the demographics of the subject should not solely influence the outcome of the model. Otherwise, the model may reinforce harmful stereotypes resulting in negative consequences for certain group identities (van Miltenburg, 2016).

Motivated by this argument, we investigate the effect of shifting the projection of gender-marking to a gender-neutral space by continued pretraining on gender-neutral multimodal data–a form of domain adaptation (Gururangan et al., 2020)–and how it reflects on task performance after fine-tuning. Figure 1 depicts an overview of our full workflow.

**Contributions** We examine whether bias amplification measured on pretrained V&L models (intrinsic bias) relates to bias amplification measured on downstream tasks (extrinsic bias). We show that a biased pretrained model might not translate into biased performance on a downstream task to a similar degree. Likewise, we measure model fairness through group disparity and show that it is not unequivocally related to bias in a model. Furthermore, we empirically present a simple, viable approach to promote fairness in V&L models: performing an extra epoch of pretraining on unbiased (gender-neutral) data reduces fine-tuning variance and group disparity on VQAv2 and retrieval tasks on the majority of models studied, without significantly compromising task performance.

We make our code publicly available to ensure reproducibility and foster future research.[1]

---

[1]http://github.com/coastalcph/gender-neutral-vl

## 2 Related Work

**Bias in language** In general, bias can be defined as "undue prejudice" (Crawford, 2017). Studies targeting language models (Kurita et al., 2019; Zhao et al., 2019) have shown that biases encoded in pretrained models (*intrinsic bias*) can be transferred to downstream applications (*extrinsic bias*), but the relationship between these biases is unclear.[2] There are several studies (Goldfarb-Tarrant et al., 2021; Delobelle et al., 2021; Kaneko et al., 2022; Cao et al., 2022; Orgad et al., 2022), showing that intrinsic bias in language models does not consistently correlate with bias measured extrinsically on a downstream task or, similarly, with empirical fairness (Shen et al., 2022; Cabello et al., 2023). Contrarily, Jin et al. (2021) observed that the effects of intrinsic bias mitigation are indeed transferable in fine-tuning language models. To the best of our knowledge, we are the first to investigate if the same holds for V&L models.

**Bias in vision & language** Prior research observed the presence of gender disparities in visual datasets like COCO (Bhargava and Forsyth, 2019; Zhao et al., 2021; Tang et al., 2021) and Flickr30k (van Miltenburg, 2016). Recent studies also revealed the presence of unwanted correlations in V&L models. Prejudicial biases found in V&L models are not only attributed to one domain, *i.e.*, vision or language, but they are compound (Wang et al., 2019), and this should be studied together. Srinivasan and Bisk (2021); Hirota et al. (2022b) and Zhou et al. (2022) show that different model architectures exhibit gender biases, often preferring to reinforce a stereotype over faithfully describing the visual scene. Bianchi et al. (2023) show the presence of stereotypes in image generation models and discuss the challenges of the compounding nature of language–vision biases. Another line of work addresses visual *contextual* bias (Choi et al., 2012; Zhu et al., 2018; Singh et al., 2020) and study a common failure of recognition models: an object fails to be recognized without its co-occurring context. So far, little work has investigated bias amplification in pretrained V&L models. Our study is among the first to cast some light on the gender bias encoded in pretrained V&L models and evaluate

---

[2]As first suggested by Goldfarb-Tarrant et al. (2021), we can broadly categorize bias into *intrinsic* and *extrinsic*. Therefore, *intrinsic metrics* are applied directly to word representations and relate bias to the geometry of the embedding space, whereas *extrinsic metrics* evaluate bias in downstream tasks.

how it translates to downstream performance.

**Gender-neutral language** Zhao et al. (2019) examine the effect of learning gender-neutral embeddings during training of static word embeddings like GloVe (Pennington et al., 2014). Sun et al. (2021) and Vanmassenhove et al. (2021) present rule-based and neural rewriting approaches to generate gender-neutral alternatives in English texts. Brandl et al. (2022) find that upstream perplexity substantially increases and downstream task performance severely drops for some tasks when gender-neutral language is used in English, Danish and Swedish. Amend et al. (2021) show that the substitution of gendered for gender-neutral terms on image captioning models poses a viable approach for reducing gender bias. In our work, we go one step beyond and investigate the effect of continued pretraining V&L models on in-domain data where gendered terms have been replaced by their gender-neutral counterparts (*e.g.*, *sister → sibling*).

## 3 Problem Formulation

We characterize the gender bias encoded in V&L models in a two-phase analysis:

*i)* Intrinsic bias: First, we investigate the bias encoded after the V&L pretraining phase.

*ii)* Extrinsic bias and task performance: Second, we fine-tune the models on common downstream tasks to further investigate how bias affects model performance.

These investigations will be performed using a set of original, pretrained models $M_D$, and models that have been further pretrained on gender-neutral data $M_D^N$ in order to mitigate any biases learned during pretraining (§4.4). We hypothesize that this bias mitigation technique will decrease both intrinsic and extrinsic biases encoded in the models.

**Data** Our analysis relies on data where the gender of the main actor of the image is known. This is, to some degree, annotated in the crowdsourced text, *e.g.*, image captions or questions.[3] Following Zhao et al. (2017) and Hendricks et al. (2018), images are labelled as 'Male' if the majority of its

captions include a word from a set of male-related tokens (*e.g.*, BOY), and no caption includes a word from the set of female-related tokens (*e.g.*, GIRL); and vice-versa for 'Female'. Images are labelled as 'Neutral' if most of the subjects are listed as gender-neutral (*e.g.*, PERSON), or if there is no majority gender mention in the texts. Finally, images are discarded from the analysis when the text mentions both male and female entities, or there are no people mentioned. This process can be applied to both pretraining data and downstream task data. See Appendix A for the complete word list.

## 4 Measuring Bias in V&L Models

### 4.1 Intrinsic Bias

When we measure the intrinsic bias of a model, we are interested in whether there are systematic differences in how phrases referring to demographic groups are encoded (Beukeboom et al., 2014). We can measure the intrinsic bias using the model's language modelling task, where the tokens related to grammatical gender are masked.[4]

Let $M_D$ be a V&L model pretrained on corpora $D$. The masked words related to grammatical gender are categorised on $N = 3$ disjoint demographic groups $A = \{\text{Male}, \text{Female}, \text{Neutral}\}$ based on reported visual appearance in the image. The gender associated with an image is considered as the ground truth (see previous section for more details). Let $g_i$ for $i \in [1, N]$ be the categorical random variable corresponding to the presence of the group $i$. We investigate the **gender–context distribution**: the co-occurrence between attributes $A_i = \{a_1, \ldots, a_{|A_i|}\}$, *e.g.*, gender terms, for a demographic group $g_i$, and contextual words $T = \{t_1, \ldots, t_T\}$, *e.g.*, objects that appear in a given text. This results in a co-occurrence matrix $C_{a,t}^{g_i}$ that captures how often pairs of attribute–context words occur in a defined context $S$, *e.g.*, an image caption in a corpus $\mathcal{C}$. Formally, for every demographic group $g_i$, over the $A_i$ attributes and $T$ objects, and all possible contexts in corpus $\mathcal{C}$

$$C_{a,t}^{g_i} = \sum_{S \in \mathcal{C}} \sum_{j=1}^{|A_i|} \sum_{k=1}^{|T|} S(a_j, t_k) \quad \text{with } i \in [1, N],$$
(1)

where $S(a_j, t_k) = 1$ if the attribute and object co-occur, zero otherwise. Based on $C_{a,t}^{g_i}$, standard

---

[3]Zhao et al. (2021) annotated samples from the COCO dataset (Lin et al., 2014) with the perceived attributes (gender and skin-tone) of the people in the images. However, their gender labels agree on 66.3% of the images compared to caption-derived annotations. To be consistent across all datasets used in our project, we will not use their human-collected annotations for analysing gender bias on COCO.

[4]We define *gender* correlations as our case study of representational bias, but note that our methodology can be extended to analyse bias with regard to any protected attribute(s).

statistical metrics like precision, recall and F1 can be computed. In addition, we will quantify the bias amplification in a given model $M_D$ to better understand the degree of bias exacerbated by the model. We use the metric presented by Wang and Russakovsky (2021), which is described in more detail in the next section.

## 4.2 Bias Amplification

We use the BiasAmp metric introduced by Wang and Russakovsky (2021), as it accounts for varying base rates of group membership and naturally decouples the direction of bias amplification: While $\text{BiasAmp}_{T \to A}$ measures the bias amplification due to the *task* influencing the protected *attribute* prediction,[5] $\text{BiasAmp}_{A \to T}$ measures the bias amplification due to the *protected attribute* influencing the task prediction. We give a concise treatise of $\text{BiasAmp}_{A \to T}$ here, and refer to Wang and Russakovsky (2021) for further details.

In our setup, the set of attributes $a \in A$ is given by $A = \{\text{Male, Female, Neutral}\}$, and the set of tasks (or objects) $t \in T$ are the most frequent nouns co-occurring with gendered terms in the training sets (see Appendix A for details). Denote by $P(T_t = 1)$ the probability that an example in the dataset belongs to class $t$. And, similarly, $P(\hat{T}_t = 1)$ the probability that an example in the dataset is labelled as class $t$ by the model. Wang and Russakovsky (2021) introduce two terms to disambiguate the direction of bias amplification. The first term, $\Delta_{at}$, quantifies the difference between the bias in the training data and the bias in model predictions.

The second term, $y_{at}$, identifies the direction of correlation of $A_a$ with $T_t$; that is, $y_{at}$ alters the sign of the $\Delta_{at}$ to correct for the fact that the bias can have two directions. Thereby,

$$\text{BiasAmp}_{A \to T} = \frac{1}{|A||T|} \sum_{\substack{a \in A \\ t \in T}} y_{at}\Delta_{at} - (1 - y_{at})\Delta_{at} \quad (2)$$

$\text{BiasAmp}_{A \to T}$ will be *positive if the model predictions amplify the prevalence of a class label $t \in T$ between groups $a \in A$ in the dataset*. For instance, bias is amplified if $A_a = \text{MALE}$ images are more likely to appear in the presence of a

$T_t = \text{SKATEBOARD}$ in the model predictions, compared to the prior distribution from the dataset. In contrast, a negative value indicates that model predictions diminish the bias present in the dataset. A value of 0 implies that the model does not amplify the bias present in the dataset. Note that this does not imply that the model predictions are unbiased.

## 4.3 Extrinsic Bias & Fairness

The second phase of our analysis measures extrinsic bias amplification: downstream performance and fairness (group disparity). A given model is fine-tuned on downstream tasks that require different reasoning skills based on the image context. We evaluate model performance with respect to the three demographic groups defined in $A$ and compare results in search of the more equitable system.

## 4.4 Gender-neutral Domain Adaptation

Motivated by the fact that models are known to acquire unwanted biases during pretraining (Hall et al., 2022), we also investigate what happens if a model $M_D$ is further pretrained for one additional epoch on gender-neutral data, with the goal of creating a more gender-neutral model $M_D^N$. We hypothesize that this may be sufficient to reduce the biases encoded in the original model. Given a dataset $D$, a new dataset $D^N$ is created by substituting gender-related tokens in the text for gender-neutral tokens. The substitution is based on a hand-crafted lexicon,[6] *e.g.*, *woman* or *man* may be substituted to *person*.[7] The new model $M_D^N$ is used for both the intrinsic and extrinsic bias evaluations.

## 5 Experimental setup

### 5.1 Models

We take the LXMERT architecture (Tan and Bansal, 2019) as a popular representative of V&L models, and build our controlled analysis on VOLTA (Bugliarello et al., 2021). VOLTA is an implementation framework that provides a fair setup for comparing V&L models pretrained under the same conditions, which enables us to compare the influence of diverse training data on representational bias. In this case, LXMERT$_{180K}$ refers to the original checkpoint and LXMERT$_{3M}$ to the model trained on CC3M (Bugliarello et al., 2021). We also study

---

[5]We do not consider gender prediction as a task per se, as gender –or any other sensitive attribute– prediction entangles a complex categorization and a moral debate (Keyes, 2018; Larson, 2017). Instead, we use a MLM task as proxy and ask the model to predict the subject of a sentence given its context.

[6]See Appendix A

[7]Note that when the pretraining data $D$ is composed of multiple corpora, we argue that domain adaptation to a non-biased space should be performed only on *clean* data, and, therefore, $|D^N| \leq |D|$.

ALBEF in two sizes and BLIP. Table 1 lists the models included in our analysis.

## 5.2 Gender-neutral Data

As a natural extension to study representational gender bias, we want to evaluate to what extent gender-neutral data helps to mitigate gender bias. Amend et al. (2021) showed that gender-neutral training might be a viable approach for reducing gender bias in image captioning models. We study its effect in more generic pretrained V&L models.

The gender-neutral pretraining data is the result of substituting terms with grammatical gender for gender-neutral equivalents, *e.g.*, "A woman walking her dog" translates into "A person walking their dog." To this end, we create a list of gender entities[8] by merging previous hand-curated lexicons used in a similar context to ours, provided by Antoniak and Mimno (2021).[9]

Starting from a pretrained checkpoint, we perform an extra epoch of pretraining. The training is done based on a linear function that increases the probability for a model to learn from gender-neutral captions. The starting rate is $p=0.15$ and, as the training progresses, the probability of getting a gender-neutral caption increases to $p=1.0$ at the last step. Note that as the probability of getting a gender-neutral caption increases, the learning rate decreases. This methodology supports our intuition that starting with a gender-neutral corpus would be too drastic for the model to adapt to, and instead cause catastrophic forgetting.

Finally, we continue pretraining the original model checkpoints for an extra epoch *without* the gender-neutral alternative (*i.e.*, $p=0.0$). The evaluation on this new checkpoint will help us to draw conclusions on longer training, as well as ensure the correct implementation of our setup.

## 5.3 Evaluation Tasks

For evaluation of downstream tasks, we report task performance and analyse group disparities. Bias amplification is reported on the validation splits.

**MLM** We follow standard practice for assessing gender bias in V&L models (Zhao et al., 2017; Hendricks et al., 2018; Wang et al., 2019; Tang et al.,

---

[8]See Appendix A for the complete list.

[9]We deliberately omit tokens like 'actor' from the list if the female (or male) equivalent is not always used (people do not always use the word 'actress' when referring to a female character). We also discard 'male' and 'female' as we suspect that they are more often used on non-human entities.

| Model ($M_D$) | Gender-neutral model ($M_D^{\text{N}}$) |
|---|---|
| $\text{LXMERT}_{180K}$ | $\text{LXMERT}_{180K}^{\text{N}}$ |
| $\text{LXMERT}_{3M}$ | $\text{LXMERT}_{3M}^{\text{N}}$ |
| $\text{ALBEF}_{4M}$ | $\text{ALBEF}_{4M}^{\text{N-COCO}}$, $\text{ALBEF}_{4M}^{\text{N-CC3M}}$ |
| $\text{ALBEF}_{14M}$ | $\text{ALBEF}_{14M}^{\text{N-COCO}}$, $\text{ALBEF}_{14M}^{\text{N-CC3M}}$ |
| $\text{BLIP}_{129M}$ | $\text{BLIP}_{129M}^{\text{N}}$ |

Table 1: Summary of the models. The subscript in the model name indicates the number of images in the pretraining set. *All* gender-neutral models are pretrained with in-domain data ($\text{LXMERT}_{180K}^{\text{N}}$ and $\text{BLIP}_{129M}^{\text{N}}$ on COCO; $\text{LXMERT}_{3M}^{\text{N}}$ on CC3M). For models with more than one gender-neutral version, the superscript indicates the dataset used for gender-neutral pretraining.

2021; Srinivasan and Bisk, 2021; Agarwal et al., 2021; Cho et al., 2022) and expose representational bias in a masked language modelling (MLM) task. The words masked are gendered terms given by the same lexicon used in §5.2. Personal pronouns (if any) are also masked to avoid leaking gender information into the model representation. For example, "A woman walking her dog" would be masked as "A [MASK] walking [MASK] dog". The image associated with each sentence is also input to the model, in a setup that reflects the pretraining conditions.

We investigate the intrinsic bias of the models as detailed in §3, *i.e.*, we look at the co-occurrence of context words (*e.g.*, car, ball) with particular word choices from the model (*e.g.*, gender words like woman, child). Previous work (Sedoc and Ungar, 2019; Antoniak and Mimno, 2021; Delobelle et al., 2021) showcases how the measure of bias can be heavily influenced by the choice of target seed words. To avoid misleading results from low frequency words, we define the set of target words to be the 100 most frequent common nouns that co-occur with the gender entities in the corresponding training data. Table 2 provides a summary of gender distribution.

To evaluate intrinsic bias, we do not look at the exact word prediction but instead consider two options to annotate the gender of the predicted word. First, we can extract and sum the probabilities of *all* male, female and gender-neutral tokens within our set to select the most probable gender entity. However, given that the distributions of tokens follows Zipf's Law, the probability mass computed for each gender group is nearly equal, yielding inconclusive results. Therefore, we use the gender category of the most probable token. Then, the bias

| | COCO | CC3M | VQAv2 | GQA | NLVR2 | F30K |
|---|---|---|---|---|---|---|
| | Image | Image | Question | Question | Sentence | Image |
| Male | 725 | 901 | 20000 | 8265 | 91 | 345 |
| Female | 363 | 945 | 9498 | 4860 | 99 | 207 |
| Neutral | 1187 | 1095 | 18549 | 4442 | 377 | 336 |
| Total | 2275 | 2941 | 48047 | 17567 | 567 | 889 |

Table 2: Gender distribution across validation splits in each dataset. Note that for COCO, this refers to the minival split in (Tan and Bansal, 2019). COCO and F30K have five captions per image. Gender was inferred from image captions for COCO, CC3M and F30K. Gender was inferred from questions in VQAv2, GQA and from the sentence given in NLVR2.

present in model predictions is measured with the statistical and bias amplification metrics presented in §4.2.

**Visual Question Answering** VQA (Antol et al., 2015) requires the model to predict an answer given an image and a question. LXMERT formulates VQA as a multi-answer classification task, and AL-BEF and BLIP treat it as a language generation task. We evaluate models on the VQAv2 (Goyal et al., 2017) and GQA datasets (Hudson and Manning, 2019), and report performance as VQA-Score and accuracy, respectively.

Bias amplification is measured on the subset of question–answer pairs targeting people. Gender is inferred from the question, considering all the gender entities presented in Appendix A. We filter any answer category whose answer does not occur with gender entities at least 50 times in the training set. Finally, numerical and yes/no question-answer pairs are also removed leaving a total of 165 answer categories in VQAv2 and 214 in GQA.

**Natural Language for Visual Reasoning** NLVR2 (Suhr et al., 2019) requires the model to predict whether a text describes a pair of images. The notion of bias amplification considered in this project would require us to manually annotate the gender from all the images to be able to extract gender-context patterns from the training data. For this reason, we only evaluate the group disparity in NLVR2 through differences in performance, reported as accuracy.

**Image–Text Retrieval** This retrieval task contains two subtasks: text-to-image retrieval (IR), where we query the model with a caption to retrieve an image, and image-to-text retrieval (TR), where we use an image to retrieve a suitable caption. We report Recall@1 on the Flickr30K (Plummer et al., 2015) benchmark. Bias amplification is measured on the subset of data targeting people. In IR, we query the model with captions that include a word from the set of male-related or female-related tokens and compare to the gender annotated in the image retrieved. In TR, we query the model with images annotated as 'Male' or 'Female' and compare to the gendered terms in the caption retrieved. Captions with gender-neutral terms are treated as a separate case to assess how often the models retrieve images from each group, yet the image retrieved could be potentially valid for any gender case. In both subtasks, we consider that the model does not amplify gender bias when the image or caption retrieved has a gender-neutral subject.

# 6 Results

## 6.1 Intrinsic Bias

We evaluate intrinsic bias in encoder-only models. Considering that bias varies as a function of the bias in a dataset, amongst other variables (Hall et al., 2022), we define our experiments with LXMERT variants as our *control setup*: the same model architecture is trained with the same hyperparameters on disjoint corpora yielding two versions of the model, $LXMERT_{180K}$ and $LXMERT_{3M}$.

**Gender-neutral pretraining mitigates gendered outputs** Figure 2 shows results for $LXMERT_{180K}$ models; complete results are in Appendix C. A model is penalised when it predicts a token from the opposite gender, but we consider a gender-neutral term as a valid output.[10] The models pretrained with gender-neutral data, have near perfect F1 performance as they learnt to predict gender-neutral tokens when their standard counterparts, $LXMERT_{180K}$ and $LXMERT_{3M}$, had low confidence on the most probable token.[11] We presume these are images where the visual appearance of the main subject is unclear. Interestingly, the trade-off between precision and recall has opposite directions for Female and Male groups *vs* Neutral in $LXMERT_{180K}$ and $LXMERT_{3M}$: the models tend to output female- and male- tokens more often than neutral-related, even when the subject in the

---

[10] Predicting a gender-neutral term shows that the model understands the depicted visual concept at the generic level.

[11] The models do not *forget* to predict gender-related tokens. $LXMERT_{180K}^{N}$ predicts ~37% of the time a word from the set of neutral-related tokens (compared to ~20% in $LXMERT_{180K}$).

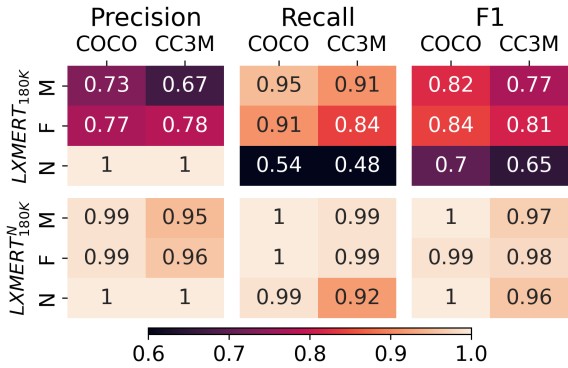

Figure 2: Statistical analysis of gender bias in MLM with gendered terms masked. Predicting a token from the gender-neutral set is always considered correct (Precision=1). Models report higher recall scores for Male (M) and Female (F) groups, showcasing the completeness of positive predictions; it is the opposite for Neutral (N) tokens.

image was annotated as gender neutral (low recall).

**Pretrained models reflect training data biases** Table 3 shows the aggregated bias amplification measured in encoder-only model variants. Our bias mitigation strategy has the same consistent behaviour across LXMERT models and evaluation data (COCO or CC3M): models tend to reflect the same degree of bias present in the data (BiasAmp$_{T \to A}$ closer to zero). ALBEF$_{14M}^{N\text{-}COCO}$ and ALBEF$_{14M}^{N\text{-}CC3M}$ models benefit from pretraining on gender-neutral data differently, as both decrease the overall bias amplification. Wang and Russakovsky (2021) caution against solely reporting the aggregated bias amplification value, as it could obscure attribute-task pairs that exhibit strong bias amplification. We report it here as a relative metric to compare the overall amplified bias between the models, and should not be considered in its own. See Appendix C for results broken down by gender.

We also investigated the equivalent to LXMERT$_{3M}^{N}$, but pretrained on gender-neutral data for a reduced number of steps to match those in LXMERT$_{180K}^{N}$. We verified that more pretraining steps on gender-neutral data equates to a reduced bias amplification in absolute terms.

## 6.2 Extrinsic Bias & Fairness

**Trade-offs in task performance** Downstream performance on the test sets is shown in Table 4. LXMERT$_{180K}$ may require more pretraining steps to converge, as we verify that the performance improvement observed in LXMERT$_{180K}^{N}$ is mainly due to the extra pretraining steps regardless of gender-neutral data. Our strategy for mitigating gender bias on pretrained models generally leads to lower task performance on NLVR2 and image retrieval, revealing a *trade-off between bias mitigation and task performance*. The same trade-off has been observed in language models (He et al., 2022; Chen et al., 2023). However, gender-neutral models report similar or even superior performance on question answering and text retrieval tasks compared to their original versions.

**Gender-neutral models consistently reduce group disparity** Group performance is depicted in Figure 3 for a subset of models and tasks. Table 7 in Appendix D shows the complete results. We observe that group disparity is consistently reduced on VQAv2 and retrieval tasks. An exception are LXMERT models, which show a minor, undesirable increase in group disparity on VQAv2, GQA and text retrieval tasks. For instance, in question-answering tasks with LXMERT, we observe a reduction in the min-max gap of 4.5 (LXMERT$_{180K}^{N}$) points in VQAv2, while the min-max gap increase in GQA is *only* of 0.4 points. Note that Tan and Bansal (2019) pretrained LXMERT$_{180K}$ on GQA train and *validation* data, which results in a very high performance (~85.0 for all groups) on the GQA validation set. We speculate that the gains in performance equality across groups could be due to a shift of the final word representations to a more equidistant vector space between gendered terms and their context. That is, the conditional probability distribution of a gendered term given its context is smoother across different demographic groups. We leave exploration of this for future work. In recent work, Feng et al. (2023) continued pretraining language models on partisan corpora and observed that these models *do* acquire (political) bias from said corpora. In our case, the continued pretraining could make the $M_D^N$ models more robust regarding gendered terms.

**Gender-neutral training reduces fine-tuning variance** Dodge et al. (2020) and Bugliarello et al. (2021) analysed the impact of random seeds in fine-tuning. We do this analysis on our control setup and observe that gender-neutral variants of LXMERT consistently report lower variance in performance on all tasks, except for NLVR2. We, however, observe a strong variance in the fine-tuning process for NLVR2 due to the random

|  | LXMERT$_{180K}$ | LXMERT$_{180K}^{N}$ | LXMERT$_{3M}$ | LXMERT$_{3M}^{N}$ | ALBEF$_{14M}$ | ALBEF$_{14M}^{N-COCO}$ | ALBEF$_{14M}^{N-CC3M}$ |
|---|---|---|---|---|---|---|---|
| COCO | -.0359 | -.0008 | -.0617 | -.0014 | -.0742 | -.0517 | -.0792 |
| CC3M | -.0346 | -.0062 | -.0007 | -.0002 | -.0182 | -.0367 | -.0570 |

Table 3: BiasAmp$_{T \to A}$ averaged over attributes (gender entities) and tasks (top-100 nouns) for LXMERT and ALBEF$_{14M}$ models. Light and dark backgrounds indicate bias amplification measured in-domain and out-of-domain data respectively. Negative values indicate an overall decrease of the bias in model's predictions.

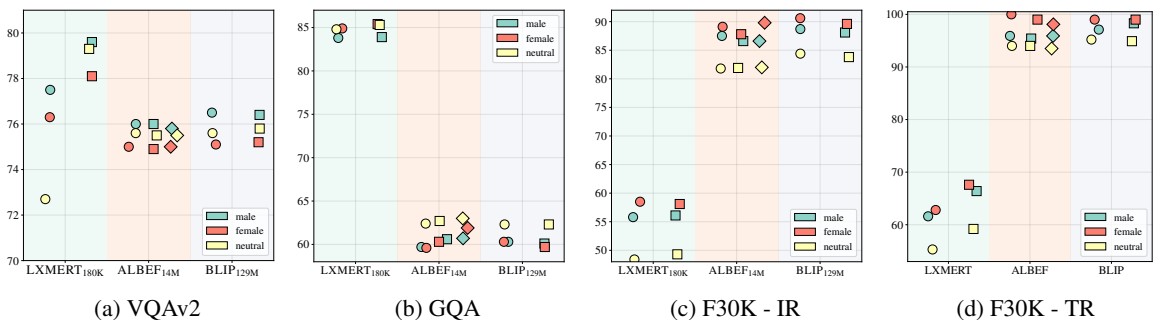

(a) VQAv2     (b) GQA     (c) F30K - IR     (d) F30K - TR

Figure 3: Validation-set results of selected models ($\circ$: LXMERT$_{180K}$, ALBEF$_{14M}$ and BLIP$_{129M}$) and their gender-neutral version ($\square$: LXMERT$_{180K}^{N}$, ALBEF$_{14M}^{N-COCO}$ and BLIP$_{129M}^{N}$, $\diamond$: ALBEF$_{14M}^{N-CC3M}$). We report VQA-accuracy in VQAv2, accuracy in GQA, and Recall@1 in F30K by gender group: male (M), female (F), and neutral (N).

|  | VQAv2 | GQA | NLVR2 | F30K | |
|---|---|---|---|---|---|
|  | test-dev | test-dev | test-P | test IR | test TR |
| LXMERT$_{180K}$ | 70.3 | **59.4** | **74.5** | 53.0 | 61.1 |
| LXMERT$_{180K}^{N}$ | **71.6** | 59.3 | **74.5** | **53.9** | **66.2** |
| LXMERT$_{3M}$ | 67.2 | 55.4 | **71.5** | **54.4** | **59.5** |
| LXMERT$_{3M}^{N}$ | **68.1** | **56.0** | 70.0 | 50.2 | 57.4 |
| ALBEF$_{4M}$ | **72.9** | **56.6** | **79.3** | **82.6** | 93.3 |
| ALBEF$_{4M}^{N-COCO}$ | **72.9** | 56.3 | 77.1 | 82.5 | 94.0 |
| ALBEF$_{4M}^{N-CC3M}$ | **72.9** | **56.6** | 78.4 | 82.4 | **94.2** |
| ALBEF$_{14M}$ | **74.4** | **58.4** | **82.4** | **85.9** | 95.1 |
| ALBEF$_{14M}^{N-COCO}$ | 74.1 | 57.3 | 52.3[12] | 85.5 | **95.4** |
| ALBEF$_{14M}^{N-CC3M}$ | 74.1 | 58.1 | 81.0 | 85.1 | 95.2 |
| BLIP$_{129M}$ | **75.3** | 58.1 | **79.7** | **87.5** | **96.7** |
| BLIP$_{129M}^{N}$ | 75.2 | **58.3** | 79.3 | 86.9 | 96.2 |

Table 4: Test results for a model $M_D$ and its gender-neutral version $M_D^N$. We report VQA-accuracy in VQAv2, accuracy in GQA and NLVR2, and Recall@1 in F30K. Results for original models computed by us.

weight initialisation of the classification layer. See Appendix E for specific results across 6 runs.

**Intrinsic & extrinsic bias are independent** We estimate bias amplification in VQA tasks by evaluating the fluctuations in models' predictions when they differ from the correct answer. Otherwise, the models are said to not amplify the bias from the data. We find that *all* model variants − $M_D$ and $M_D^N$ − reduce the gender bias across tasks.

However, contrary to what we observed in pretrained models (Table 3), there is no evidence that the gender-neutral pretraining influenced positively (nor negatively) the extrinsic bias of the models: it depends on the model, downstream task and gender group (see Appendix E for results on BiasAmp$_{A \to T}$ fine-tuning variance). Figure 4 displays BiasAmp$_{A \to T}$ broken-down by gender category measured on GQA for a subset of models. Whereas the degree of bias amplification is fairly consistent between a model $M_D$ and $M_D^N$ in VQAv2 (see Appendix D), there is higher variance in GQA: ALBEF$_{14M}^{N-COCO}$ reduces the bias amplification compared to ALBEF$_{14M}$, but we observe the opposite effect on BLIP$^{N-COCO}$.

In retrieval tasks, we look into models' behavior when querying them with neutral instances. Regardless of the degree of intrinsic bias in the model, models exhibit the same trend: in IR, all models mostly retrieve images labeled as 'Neutral', but twice as much 'Male' images as 'Female'. We find similar results for TR, *i.e.*, query images whose main actor is defined as Neutral, but, in this scenario, only half of the captions retrieved relate to people. See Appendix D for detailed results.

## 7 Conclusion

This paper presented a comprehensive analysis of gender bias amplification and fairness of encoder-only and encoder-decoder V&L models. The in-

---

[12]This result is inexplicably low, despite fifteen attempts at fine-tuning with different random seeds. We saw similar instabilities when fine-tuning the released LXMERT models, but we found seeds that gave above-chance accuracy.

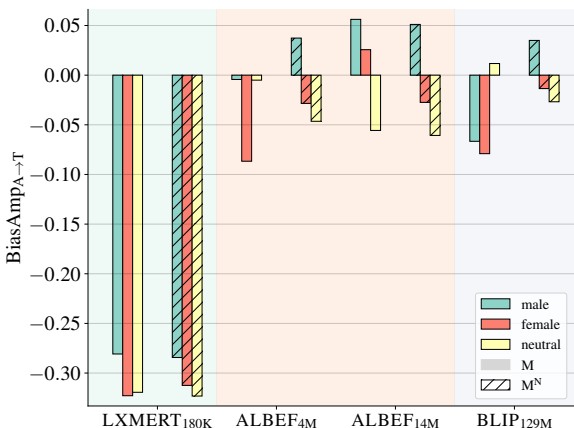

Figure 4: Bias amplification measured on question-answering (GQA) broken down by gender group. $M^{\mathrm{N}}$ are gender-neutral pretrained on COCO.

trinsic bias analysis shows consistent results – in terms of bias mitigation – in models trained on gender-neutral data, even if these models reflect biases present in data instead of diminishing them (as we observed with LXMERT). In line with previous findings in language models (Goldfarb-Tarrant et al., 2021; Kaneko et al., 2022; Orgad et al., 2022), intrinsic bias in V&L models does not necessarily transfer to extrinsic bias on downstream tasks. Similarly, we find that the bias in a model and its empirical fairness –group disparity on task performance– are in fact independent matters, which is in line with the NLP literature (Shen et al., 2022; Cabello et al., 2023). Intrinsic bias can potentially reinforce harmful biases, but these may not impact the treatment of groups (or individuals) on downstream tasks. We believe that bias and fairness should always be carefully evaluated as separate matters. One of they key findings of our work is that the extra pretraining steps on gender-neutral data are beneficial to reduce the group disparity in every model architecture tested on VQAv2, and in the majority of models for both retrieval tasks. Crucially, there is no penalty to pay for this fair outcome: the overall task performance of gender-neutral models is similar or better than their original versions.

## Limitations

The framework to characterize gender bias in V&L presented in this study is general and extensible to analyse other forms of bias in multimodal models. We consider three base architectures to settle on the implementation. However, our work would benefit from analyzing a wider range of models.

Studying the effects of gender-neutral pretraining on V&L models with a frozen language model, such as ClipCap (Mokady et al., 2021) and BLIP-2 (Li et al., 2023), is left as future work.

Due to computational limitations, we restricted most of our analysis to single runs. We perform a first analysis across multiple random seeds for LXMERT models in Appendix E. There, we notice that gender-neutral models seem to have lower variance after fine-tuning. Yet, the cross-seed performance of a given model can fluctuate considerably for some tasks (e.g., NLVR2), corroborating previous findings from Bugliarello et al. (2021). Likewise, bias amplification, along with other fairness metrics like group disparity, often fluctuates across runs. We report bias amplification variance in fine-tuning of LXMERT models, but the absence of confidence intervals for all models and tasks – due to the same reason stated above– should be considered. We hope to motivate future work to address this issue.

Moreover, despite the existence of multilingual multimodal datasets (Elliott et al., 2016; Liu et al., 2021; Bugliarello et al., 2022, inter-alia), our experimental setup is limited to English datasets and models. Studies of (gender) bias using only English data are not complete and might yield inaccurate conclusions, albeit overcoming the structural pervasiveness of gender specifications in grammatical gender languages such us German or Spanish is not trivial (Gabriel et al., 2018). Likewise, our work considers a single dimension of social bias (gender). Further research on analyzing social biases on V&L models should account for intersectionality: how different social dimensions, e.g., gender and race, can intersect and compound in ways that can potentially impact model performance on most disfavoured groups, e.g., Black Women as discussed in Crenshaw (1989).

## Ethics Statement

The models and datasets used in this study are publicly available, and we strictly follow the ethical implications of previous research related to the data sources. Our work is based on sensitive information such as gender, based on reported visual appearance in the image captions. We would like to emphasize that we are not categorizing biological sex or gender identity, but rather using the given image captions as proxies to the outward gender appearance.

## Acknowledgments

We are grateful to Benjamin Rotendahl and Rita Ramos for initial discussions about data and evaluation. We also thank members of CoAStaL and LAMP groups for their valuable feedback. Laura Cabello is funded by the Novo Nordisk Foundation (grant NNF 20SA0066568). 🇪🇺 Emanuele Bugliarello is supported by the funding from the European Union's Horizon 2020 research and innovation programme under the Marie Skłodowska-Curie grant agreement No 801199. Stephanie Brandl is funded by the European Union under the Grant Agreement no. 10106555, FairER. Views and opinions expressed are those of the author(s) only and do not necessarily reflect those of the European Union or European Research Executive Agency (REA). Neither the European Union nor REA can be held responsible for them. This work was supported by a research grant (VIL53122) from VILLUM FONDEN.

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

## A  Seed words

**Gender terms**

- **Female:** aunt, bride, businesswoman, daughter, daughters, fiancee, fiancée, gal, gals, girl, girlfriend, girls, grandmother, her, herself, lady, landlady, mama, mom, mother, queen, she, sister, sisters, spokeswoman, wife, woman, women, womens.

- **Male:** boy, boyfriend, boys, brother, brothers, businessman, dad, dude, dudes, father, fiance, fiancé, gentleman, grandfather, groom, guy, he, him, himself, his, husband, king, landlord, man, men, mens, papa, son, sons, spokesman, uncle.

- **Neutral:** businessperson, child, childs, grandparent, kid, kids, landlord, monarch, newlywed, parent, partner, pbling, people, person, sibling, siblings, someone, spokesperson, spouse, their, them, themself, they.

**Gender-neutral mappings**  Using the gender terms listed above, we generate mappings from male and female to neutral terms: see Table 5 for details. These mappings are used to continue pre-training on gender-neutral (debiased) data as explained in §5.2.

**Objects**  List of top-100 most frequent nouns co-occurring with gender terms in the training split in COCO (Lin et al., 2014) and Conceptual Captions (CC3M) (Sharma et al., 2018).

- **COCO:** tennis, group, street, baseball, table, dog, front, ball, player, field, snow, game, beach, horse, skateboard, umbrella, water, phone, kite, hand, top, board, ski, couple, motorcycle, food, elephant, People, picture, pizza, surfboard, room, shirt, bench, wave, frisbee, court, park, air, cake, bed, laptop, train, cell, racket, bat, bus, kitchen, plate, glass, ocean, side, grass, giraffe, building, city, skier, road, car, suit, trick, cat, tie, tree, bike, photo, boat, hat, slope, baby, area, sign, chair, sidewalk, computer, hill, head, surfer, mountain, video, skateboarder, soccer, truck, banana, couch, camera, skate, crowd, lot, snowboard, background, wine, bear, day, back, luggage, cow, living, fence, ramp.

- **CC3M:** player, team, actor, football, game, artist, hand, day, match, background, dress,

| Male | Female | Neutral |
|------|--------|---------|
| boy | girl | child |
| boyfriend | girlfriend | partner |
| boys | girls | kids |
| brother | sister | sibling |
| brothers | sisters | siblings |
| businessman | businesswoman | businessperson |
| dad | mom | parent |
| dude | gal | person |
| dudes | gals | people |
| father | mother | parent |
| fiance | fiancee | partner |
| fiancé | fiancée | partner |
| gentleman | lady | person |
| grandfather | grandmother | grandparent |
| groom | bride | newlywed |
| guy | gal | person |
| he | she | they |
| him | her | them |
| himself | herself | themself |
| his | her | their |
| husband | wife | spouse |
| king | queen | monarch |
| landlord | landlady | landlord |
| man | woman | person |
| men | women | someone |
| mens | womens | people |
| papa | mama | parent |
| son | daughter | kid |
| sons | daughters | childs |
| spokesman | spokeswoman | spokesperson |
| uncle | aunt | pbling |

Table 5: Gender-neutral mappings used for continual pre-training in gender-neutral data as described in §5.2.

beach, car, photo, dog, event, street, home, ball, wedding, family, city, film, time, tree, award, goal, hair, front, night, water, baby, business, illustration, politician, sport, show, way, portrait, face, book, premiere, fan, room, head, friend, year, athlete, park, house, fashion, soccer, character, flower, country, style, field, side, party, festival, picture, stage, rock, eye, couple, world, shirt, vector, camera, pop, tv, ceremony, hat, glass, snow, horse, school, road, phone, arm, art, window, crowd, sea, table, part, boat, suit, basketball, model, top, birthday, star, student, view, tennis, smile, wall, celebrity, baseball.

## B  Models

In this section, we provide an overview on the models we use in our evaluation. We refer to their original work for more details.

**LXMERT** (Tan and Bansal, 2019) is a cross-modal architecture pretrained to learn vision-and-language representations. It consists of three Transformer (Vaswani et al., 2017) encoders, where visual and language inputs are encoded separately in two independent stacks of Transformer layers before feeding them into the cross-modality encoder. The cross-modality encoder uses bi-directional cross attention to exchange information and align the entities across the two modalities. LXMERT is trained with four objectives: masked language modelling (MLM), masked object prediction, image–text matching (ITM) and image question answering.

Similar to LXMERT, **ALBEF** (Li et al., 2021) is a dual-stream encoder (Bugliarello et al., 2021) that first learns separate visual and textual embeddings using Transformer-based image and text encoders; and then fuses them in a cross-modal Transformer using image–text contrastive loss (ITC), which enables a more grounded vision and language representation learning. The model is pretrained with two other objectives: masked language modelling (MLM) and image–text matching (ITM) on the multimodal encoder. Unlike LXMERT, ALBEF does not rely on image features extracted from an off-the-shelf object detector, but directly feeds the raw image into a Vision Transformer (Dosovitskiy et al., 2021)

**BLIP** (Li et al., 2022) is a versatile model based on a multimodal mixture of encoder–decoder network, that can be applied to a wide range of downstream tasks. The authors introduce a novel bootstrapping method to generate synthetic captions and remove noisy pairs from large-scale web data. Unlike LXMERT and ALBEF, BLIP is trained with an autoregressive language modelling objective that allows the generation of coherent captions given an image. The model is also pretrained using the unimodal image–text contrastive loss (ITC) and the cross-modal image–text matching (ITM) loss used by ALBEF.

## C  Bias in Pretrained Models

**Intrinsic bias**  Figure 5 complements Figure 2 from the main paper showing statistical results measured on the intrinsic bias analysis in our *control*

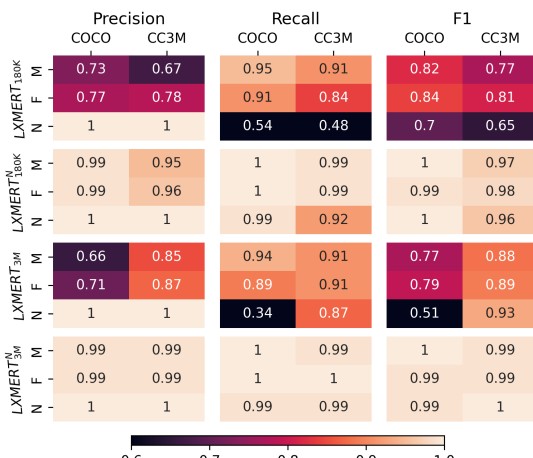

Figure 5: Statistical analysis of gender bias found through masked language modelling with gendered terms masked. Prediction of a token from the gender-neutral set is always considered correct (Precision=1). Models report higher recall scores for Male (M) and Female (F) groups, showcasing the completeness of positive predictions, whereas it is the opposite for Neutral-related (N) tokens.

*setup*.

**MLM experiment broken down by gender**  Table 6 provides a more granular look at which gender groups are actually amplifying/decreasing the bias in the pretrained models.

## D  Bias & Fairness in Downstream Tasks

**Extrinsic Bias**  The following graphs complement results shown in § 6.2 for bias amplification measured on downstream tasks: Figure 6 shows results on GQA; Figure 7 shows results on VQAv2; Figure 8 and Figure 9 show the bias revealed on image–text retrieval tasks when querying the models with a gender-neutral caption (or image), respectively.

**Task performance & Fairness**  We present granular results on task performance in validation in Table 7 and group disparity, defined as the min-max difference between group performance ($\Delta$).

|  | COCO | | |
| --- | --- | --- | --- |
|  | Male | Female | Neutral |
| LXMERT$_{180K}$ | -.0295 | -.0048 | -.0733 |
| LXMERT$_{180K}^{N}$ | -.0004 | -.0008 | -.0014 |
| LXMERT$_{3M}$ | -.0577 | -.0230 | -.1062 |
| LXMERT$_{3M}^{N}$ | -.0014 | +.0001 | -.0028 |
| LXMERT$_{180K}^{N-SC}$ | -.0082 | -.0009 | -.0109 |
| ALBEF$_{4M}$ | -.1006 | -.0517 | -.1083 |
| ALBEF$_{4M}^{N-COCO}$ | -.0748 | -.1293 | -.1529 |
| ALBEF$_{4M}^{N-CC3M}$ | -.0754 | -.0337 | -.1073 |
| ALBEF$_{14M}$ | -.0418 | -.1146 | -.0663 |
| ALBEF$_{14M}^{N-COCO}$ | -.0559 | -.0169 | -.0824 |
| ALBEF$_{14M}^{N-CC3M}$ | -.0556 | -.0983 | -.0837 |

|  | CC3M | | |
| --- | --- | --- | --- |
|  | Male | Female | Neutral |
| LXMERT$_{180K}$ | -.0281 | -.0276 | -.0482 |
| LXMERT$_{180K}^{N}$ | +.0008 | -.0081 | -.0113 |
| LXMERT$_{3M}$ | -.0043 | -.0030 | +.0055 |
| LXMERT$_{3M}^{N}$ | +.0002 | +.0004 | -.0011 |
| LXMERT$_{180K}^{N-SC}$ | -.0011 | +.0003 | -.0012 |
| ALBEF$_{4M}$ | -.0473 | -.0569 | -.0422 |
| ALBEF$_{4M}^{N-COCO}$ | -.0329 | -.0514 | -.0152 |
| ALBEF$_{4M}^{N-CC3M}$ | -.0295 | -.0497 | -.0313 |
| ALBEF$_{14M}$ | +.0159 | -.0642 | -.0062 |
| ALBEF$_{14M}^{N-COCO}$ | -.0290 | -.0250 | -.0561 |
| ALBEF$_{14M}^{N-CC3M}$ | -.0535 | -.0641 | -.0534 |

Table 6: BiasAmp$_{T \to A}$ (BA.) per gender group, averaged over tasks (top-100 nouns) for LXMERT and AL-BEF models, evaluated on validation splits on COCO (top) and CC3M (bottom). Light and dark backgrounds indicate bias amplification measured within in-domain and out-of-domain data respectively. A model amplifies the bias in the dataset if the value is positive. A negative value indicates an overall decrease of the bias in model's predictions.

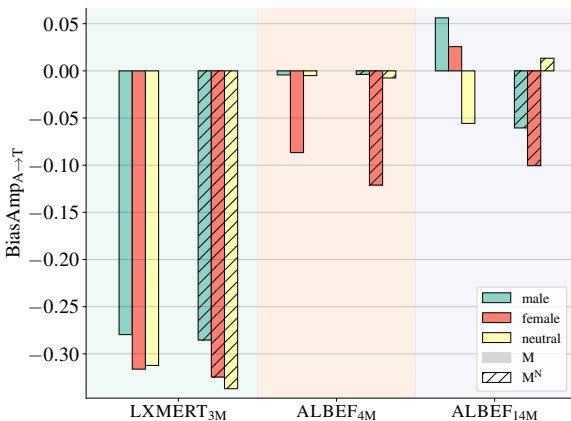

Figure 6: Bias amplification measured on question-answering (GQA) broken down by gender group. $M^N$ are gender-neutral pretrained on CC3M.

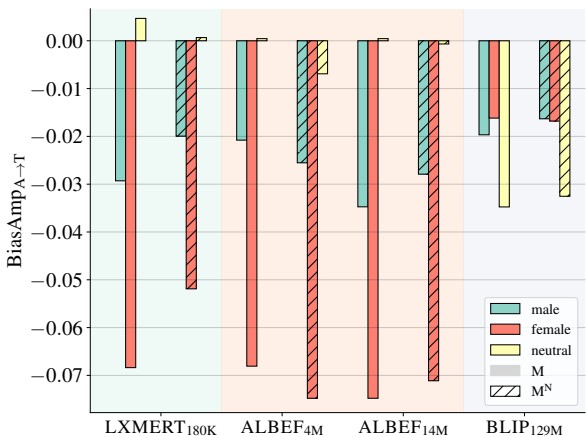

(a) $M^N$ are gender-neutral models pretrained on COCO.

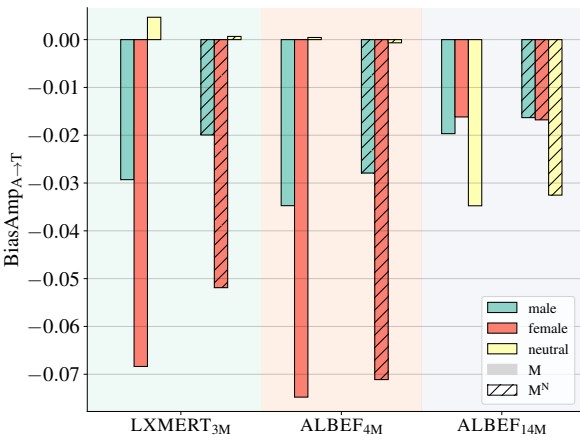

(b) $M^N$ are gender-neutral models pretrained on COCO.

Figure 7: Bias amplification measured on question-answering (VQAv2) broken down by gender group.

| | | VQAv2 | | GQA | | NLVR2 | | F30K | | | |
|---|---|---|---|---|---|---|---|---|---|---|---|
| | | Acc. | $\Delta(\downarrow)$ | Acc. | $\Delta(\downarrow)$ | Acc. | $\Delta(\downarrow)$ | r@1 IR | $\Delta(\downarrow)$ | r@1 TR | $\Delta(\downarrow)$ |
| LXMERT$_{180K}$ | M | 77.5 | | 83.8 | | 81.9 | | 55.8 | | 61.6 | |
| | F | 76.3 | 4.8 | 84.9 | **1.1** | 75.0 | 6.9 | 58.5 | 10.1 | 62.8 | **7.5** |
| | N | 72.7 | | 84.8 | | 81.1 | | 48.4 | | 55.3 | |
| LXMERT$_{180K}^{N}$ | M | 79.6 | | 83.9 | | 79.3 | | 56.1 | | 66.4 | |
| | F | 78.1 | **1.5** | 85.4 | 1.5 | 74.2 | **6.0** | 58.1 | **8.8** | 67.6 | 8.4 |
| | N | 79.3 | | 85.3 | | 80.2 | | 49.3 | | 59.2 | |
| LXMERT$_{3M}$ | M | 68.4 | | 63.7 | | 72.3 | | 56.0 | | 63.0 | |
| | F | 66.5 | 6.0 | 65.6 | **1.9** | 64.0 | 14.8 | 59.4 | 8.2 | 63.7 | **8.7** |
| | N | 62.4 | | 64.5 | | 78.8 | | 51.2 | | 55.0 | |
| LXMERT$_{3M}^{N}$ | M | 70.3 | | 64.6 | | 79.3 | | 51.4 | | 56.2 | |
| | F | 67.9 | **2.4** | 66.8 | 2.3 | 67.7 | **11.6** | 52.3 | **5.0** | 61.4 | 9.0 |
| | N | 70.1 | | 64.5 | | 78.8 | | 47.3 | | 52.4 | |
| ALBEF$_{4M}$ | M | 75.1 | | 60.0 | | 87.9 | | 83.1 | | 94.5 | |
| | F | 73.7 | 1.4 | 61.5 | 2.5 | 76.8 | 11.1 | 87.9 | 10.1 | 98.1 | 7.0 |
| | N | 74.3 | | 62.5 | | 79.6 | | 77.8 | | 91.1 | |
| ALBEF$_{4M}^{N\text{-}COCO}$ | M | 75.0 | | 60.5 | | 85.7 | | 83.7 | | 94.2 | |
| | F | 73.6 | 1.4 | 60.7 | **1.6** | 75.8 | 9.9 | 87.2 | 10.1 | 96.6 | **5.2** |
| | N | 74.0 | | 62.1 | | 79.8 | | 77.1 | | 91.4 | |
| ALBEF$_{4M}^{N\text{-}CC3M}$ | M | 75.0 | | 61.0 | | 84.6 | | 82.2 | | 94.8 | |
| | F | 73.7 | **1.3** | 60.7 | 2.0 | 74.8 | **9.8** | 87.1 | **9.2** | 97.6 | 7.7 |
| | N | 74.5 | | 62.7 | | 79.3 | | 77.9 | | 89.9 | |
| ALBEF$_{14M}$ | M | 76.0 | | 59.7 | | 86.8 | | 87.5 | | 95.9 | |
| | F | 75.0 | 1.0 | 59.6 | 2.8 | 79.8 | **7.0** | 89.1 | 7.3 | 100.0 | 6.0 |
| | N | 75.6 | | 62.4 | | 81.2 | | 81.8 | | 94.0 | |
| ALBEF$_{14M}^{N\text{-}COCO}$ | M | 76.0 | | 60.6 | | 60.4 | | 86.6 | | 95.4 | |
| | F | 74.9 | 1.1 | 60.3 | 2.4 | 52.5 | 7.9 | 87.8 | **5.9** | 99.0 | 5.0 |
| | N | 75.5 | | 62.7 | | 57.6 | | 81.9 | | 94.0 | |
| ALBEF$_{14M}^{N\text{-}CC3M}$ | M | 75.8 | | 60.7 | | 87.9 | | 86.6 | | 95.9 | |
| | F | 75.0 | **0.8** | 61.9 | **2.3** | 77.8 | 10.1 | 89.8 | 7.8 | 98.1 | **4.6** |
| | N | 75.5 | | 63.0 | | 81.7 | | 82.0 | | 93.5 | |
| BLIP$_{129M}$ | M | 76.5 | | 60.3 | | 82.4 | | 88.7 | | 97.1 | |
| | F | 75.1 | 1.4 | 60.3 | **2.0** | 77.8 | **6.0** | 90.6 | 6.2 | 99.0 | **3.8** |
| | N | 75.6 | | 62.3 | | 83.8 | | 84.4 | | 95.2 | |
| BLIP$_{129M}^{N}$ | M | 76.4 | | 60.1 | | 84.6 | | 88.1 | | 98.3 | |
| | F | 75.2 | **1.2** | 59.7 | 2.6 | 73.7 | 10.9 | 89.6 | **5.8** | 99.0 | 4.1 |
| | N | 75.8 | | 62.3 | | 80.9 | | 83.8 | | 94.9 | |

Table 7: Validation results per group: male (M), female (F), and neutral (N). We report VQA-accuracy in VQAv2, accuracy in GQA and NLVR2, recall@1 in F30k and group disparity ($\Delta$) across tasks. Lower $\Delta$ is better.

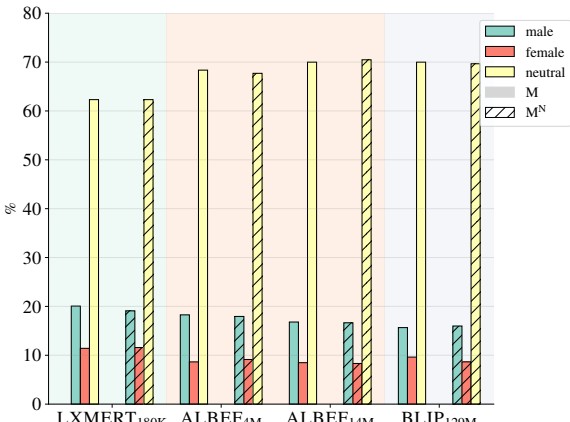

(a) IR - $M^N$ are gender-neutral models pretrained on COCO.

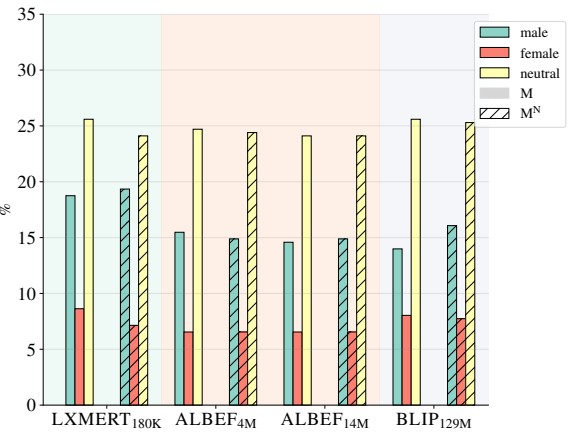

(a) TR - $M^N$ are gender-neutral models pretrained on COCO.

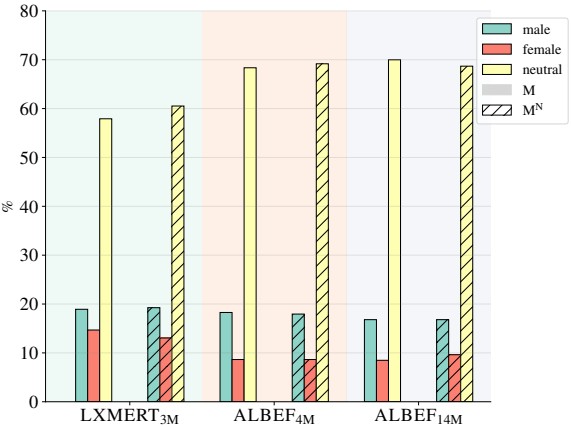

(b) IR - $M^N$ are gender-neutral models pretrained on CC3M.

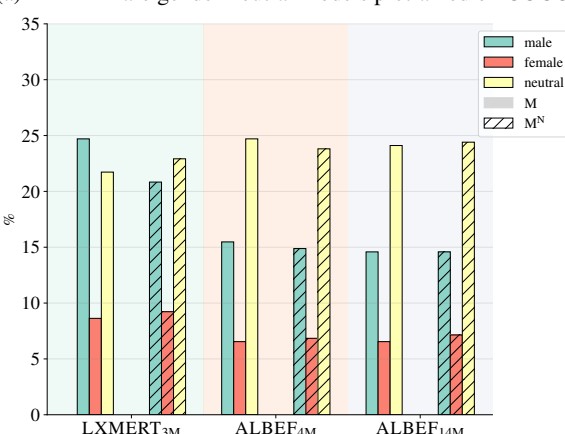

(b) TR - $M^N$ are gender-neutral models pretrained on CC3M.

Figure 8: Extrinsic bias measured on text-to-image retrieval (IR) on Flickr30K. Bias is measured as the percentage of images retrieved from each group when querying the models with a gender-neutral caption.

Figure 9: Extrinsic bias measured on image-to-text retrieval (TR) (c)-(d) on Flickr30K. Bias is measured as the percentage of captions retrieved from each group when querying the models with a gender-neutral image.

# E   Variance in fine-tuning

Table 8 shows the mean and standard deviation in bias amplification when fine-tuning LXMERT models with different random seeds. The variance is due to random initialization. In line with what we observed in §6.2, there is no clear trend when comparing a model M with it's gender-neutral pre-training counterpart, $M_D^N$.

|  |  | VQAv2 mean± std | GQA mean ± std |
|---|---|---|---|
| LXMERT$_{180K}$ | male | -.0311±.0057 | -.0192±.0071 |
|  | female | -.0497±.0053 | -.0477±.0088 |
|  | neutral | +.0020±.0030 | -.0252±.0089 |
| LXMERT$_{180K}^N$ | male | -.0301±.0031 | -.0227±.0036 |
|  | female | -.0538±.0034 | -.0528±.0106 |
|  | neutral | +.0007±.0014 | -.0245±.0042 |
| LXMERT$_{3M}$ | male | -.0169±.0054 | -.0254±.0135 |
|  | female | -.0667±.0041 | -.0935±.0110 |
|  | neutral | -.0157±.0056 | -.0269±.0069 |
| LXMERT$_{3M}^N$ | male | -.0164±.0038 | -.0188±.0060 |
|  | female | -.0634±.0046 | -.0971±.0131 |
|  | neutral | -.0183±.0035 | -.0194±.0109 |

Table 8: BiasAmp$_{A\rightarrow T}$ fine-tuning variance of LXMERT models across question answering tasks. Each model is fine-tuned 6 times on each task. We report average VQA-accuracy in VQAv2 and average accuracy in GQA, together with its standard deviation.

Figure 10 shows violin plots of the distribution of results when fine-tuning LXMERT models with different random seeds. The variance is due to random initialization. Gender-neutral models reveal lower standard deviation across tasks. This finding reveals one of the benefits to perform extra steps of pretraining on gender-neutral data: to reduce variance in downstream performance. This observation aligns with the NLP literature showing that biases in a model are independent from model performance (Cabello et al., 2023).

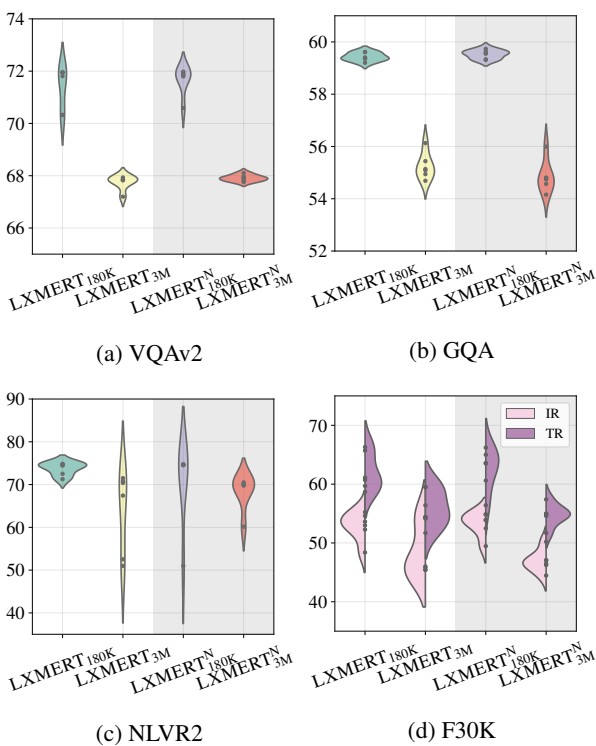

(a) VQAv2

(b) GQA

(c) NLVR2

(d) F30K

Figure 10: Fine-tuning variance of LXMERT models across tasks. On the left with white background, original models ($M_D$). On the right with darker background, models after gender-neutral pretraining ($M_D^N$). Each model is fine-tuned 6 times on each task. The dots represent the experimental observations. We report average VQA-accuracy in VQAv2, accuracy in GQA and NLVR2, and recall@1 in F30k.