# OpenReview forum: "Evaluating Bias and Fairness in Gender-Neutral Pretrained Vision-and-Language Models"
_EMNLP/2023/Conference — EMNLP 2023 Main_

### Official Review · Reviewer_cgUh · 2023-08-04

**Soundness:** 4

**Excitement:**

3: Ambivalent: It has merits (e.g., it reports state-of-the-art results, the idea is nice), but there are key weaknesses (e.g., it describes incremental work), and it can significantly benefit from another round of revision. However, I won't object to accepting it if my co-reviewers champion it.

**Paper Topic And Main Contributions:**

This paper extensively evaluates both the intrinsic bias and extrinsic bias in V&L models and examines the effectiveness of fine-tuning over gender-neutral data. Experiments show that V&L models suffer from bias, and there is no clear causal link between intrinsic bias and extrinsic bias. Moreover, gender-neutral fine-tuning could improve fairness.

**Reasons To Accept:**

- The paper is well-written and easy to follow.
- The paper provides a comprehensive literature review of related work.
- Experimental settings are clearly introduced.

**Reasons To Reject:**

- Similar research has been done for vision or language models separately in previous work, making the findings less impressive. For example, as has already been cited in the paper, Wang et al. (2019) investigated bias in vision, Zhao et al. (2019) evaluated bias in language, and Shen et al. (2022) investigated the relationship between intrinsic bias and extrinsic bias. It is not clear enough why we expect that a V&L model should differ from a V or L model in terms of bias and fairness, i.e., the motivation of this paper. As for the findings in this paper, such as 'gender-neutral pretraining mitigates gendered outputs', 'pretrained models reflect training data biases', and 'trade-offs in task performance' are consistent as findings for V or L models in the previous, which are not surprising.

**Reproducibility:**

4: Could mostly reproduce the results, but there may be some variation because of sample variance or minor variations in their interpretation of the protocol or method.

**Reviewer Confidence:**

3: Pretty sure, but there's a chance I missed something. Although I have a good feel for this area in general, I did not carefully check the paper's details, e.g., the math, experimental design, or novelty.

---

> ### Author Rebuttal · Authors · 2023-08-28
>
> Thank you for your review and valuable feedback. We address your concerns below.
>
> > Similar research has been done for vision or language models separately in previous work, making the findings less impressive. [...] It is not clear enough why we expect that a V&L model should differ from a V or L model in terms of bias and fairness, i.e., the motivation of this paper. [...]
>
> While it is true that similar research has been done independently in vision or language models, little work has addressed the presence of social biases and/or fairness in V&L models. Prejudicial biases found in V&L are not only attributed to one domain, but they are compound and need to be studied jointly. For example, [1] studied how within- and cross-modality gender biases are expressed using a set of template-based data, and demonstrated that V&L models tend to reinforce a stereotype over faithfully describing the visual scene. In the field of image generation, [2] demonstrates the presence of stereotypes in image generation models and discuss the challenges of the compounding nature of language-vision biases.
>
> References:
>
> [1] Tianlu Wang, Jieyu Zhao, Mark Yatskar, Kai-Wei Chang, and Vicente Ordonez. 2019. Balanced datasets are not enough: Estimating and mitigating gender bias in deep image representations.
> [2] Federico Bianchi, Pratyusha Kalluri, Esin Durmus, Faisal Ladhak, Myra Cheng, Debora Nozza, Tatsunori Hashimoto, Dan Jurafsky, James Zou, and Aylin Caliskan. 2023. Easily Accessible Text-to-Image Generation Amplifies Demographic Stereotypes at Large Scale.

---

### Official Review · Reviewer_xmLY · 2023-08-08

**Soundness:** 3

**Excitement:**

3: Ambivalent: It has merits (e.g., it reports state-of-the-art results, the idea is nice), but there are key weaknesses (e.g., it describes incremental work), and it can significantly benefit from another round of revision. However, I won't object to accepting it if my co-reviewers champion it.

**Paper Topic And Main Contributions:**

This paper assesses gender bias and fairness in pretrained vision & language models. The findings suggest that there is no causal link between bias amplification during pretraining and after fine-tuning. Furthermore, the study investigates whether pretrained models can mitigate group disparities when fine-tuned on gender-neutral datasets.

**Reasons To Accept:**

1. The paper introduces and clearly defines two types of gender biases: intrinsic bias and extrinsic bias.
2. The case study evaluates three vision and language models, providing valuable insights.
3. The limitation section is well-written, effectively highlighting most of the limitations of this paper.

**Reasons To Reject:**

1. The paper's novelty is not convincing, as it neither introduces new bias metrics nor proposes novel bias-mitigation methods.
2. Several important notations lack clarity. For instance, the definition of g_i in the second paragraph of 4.1 Intrinsic Bias needs further explanation. Additionally, the meaning of T in the second paragraph of 4.2 Bias Amplification is not well-defined. It is also unclear what specific tasks or classes are being referred to.
3. The paper lacks discussion regarding the reasons behind the observed benefits of "extra pretraining steps on gender-neutral data for reducing group disparity in all model architectures tested on VQAv2, and in the majority of models for both retrieval tasks."

**Reproducibility:**

3: Could reproduce the results with some difficulty. The settings of parameters are underspecified or subjectively determined; the training/evaluation data are not widely available.

**Reviewer Confidence:**

4: Quite sure. I tried to check the important points carefully. It's unlikely, though conceivable, that I missed something that should affect my ratings.

---

> ### Author Rebuttal · Authors · 2023-08-28
>
> Thank you for your thoughtful review and valuable feedback. We address your concerns below.
>
> > The paper's novelty is not convincing, as it neither introduces new bias metrics nor proposes novel bias-mitigation methods
>
> We acknowledge that our paper does not introduce a new bias metric nor a novel bias-mitigation method. Instead, the novelty lies in leveraging the use of common bias and fairness metrics in language models to understand the interaction between the two, bias and fairness, in V&L models. This is understudied in the era of large-scale pretrained multimodal models and it hardly hurts the scientific discourse to apply methodologies to different domains. In addition, we complement this analysis with a comprehensive study on the effects of continued pretraining on in-domain debiased data (gender-neutral data) which “shows a viable solution to existing biases”, as mentioned by Reviewer mmdc. Finally, Reviewer HKVB also highlights that we are “the first to investigate whether biases encoded in pretrained models can be transferred to downstream applications or not on V&L models”.
>
> > Several important notations lack clarity
>
> Thank you for pointing out unclear notations used in the paper; we hope that the following explanations will improve the clarity of our writing in the final copy.
>
> We will include further explanation regarding g_i (section 4.1) to clarify the definition given in lines 218–220. More concretely, g_i denotes the presence of one of the 3 demographic groups considered in the analysis (male, female, neutral). To make this clearer, we could rename g_i to ease the connection with the set of demographic groups defined in L214.
>
> Regarding the meaning of T in the second paragraph of section 4.2, it refers to the same T defined earlier in section 4.1; similarly, the specific tasks (or classes) that we refer to are the same defined in section 4.1. The specific words used for these tasks are listed in Appendix A. We will add a footnote to better reflect this association.
>
> > The paper lacks discussion regarding the reasons behind the observed benefits of "extra pretraining steps on gender-neutral data for reducing group disparity in all model architectures tested on VQAv2, and in the majority of models for both retrieval tasks."
>
> We agree the paper would benefit from a more extensive discussion of our findings. The effects of gender-neutral pretraining on the models’ group disparity are shown in Figure 3 in the main paper, and ​​Table 7 in Appendix 7. These results are discussed in L518–534. It is true that the current version of the paper does not elaborate much on the specific reason behind the benefits observed in terms of disparity reduction. Our intuition is that this could be due to shifting the final word representations to a more equidistant vector space between gendered terms and their context: the conditional probability distribution of a gendered term given its context is smoother across different demographic groups. This could make the pretrained model more robust before being fine-tuned. Our intuition goes in line with studies in language models such as [1]. In [1], authors continue pretraining LM on partisan data to study the political leanings of the models: they observed that LMs do acquire (political) bias from pretraining corpora, yet “left-leaning corpora generally resulted in a left/liberal shift [...]”.
>
> References:
>
> [1] Shangbin Feng, Chan Young Park, Yuhan Liu, Yulia Tsvetkov. 2023. From Pretraining Data to Language Models to Downstream Tasks: Tracking the Trails of Political Biases Leading to Unfair NLP Models.

---

### Official Review · Reviewer_HKVB · 2023-08-11

**Soundness:** 4

**Excitement:**

4: Strong: This paper deepens the understanding of some phenomenon or lowers the barriers to an existing research direction.

**Paper Topic And Main Contributions:**

This paper investigates bias amplification in pre-trained and fine-tuned vision-and-language models across three families. The study examines how intrinsic bias in pre-trained models relates to extrinsic bias in downstream tasks. Results suggest that a biased pre-trained model may not always result in biased performance in downstream tasks to the same extent. The study also shows that additional pre-training on gender-neutral data can reduce group disparities without significantly impacting task performance.

**Questions For The Authors:**

1. It is exciting to see the finding that there is no evidence that the gender-neutral pertaining influenced positively (nor negatively) the extrinsic bias of the models. However, I am confused about how to summarize the results from Appendix E and Figure 4. Can you provide more illustrations or a clearer explanation?

**Reasons To Accept:**

1. This paper is the first to investigate whether the biases encoded in pretrained models can be transferred to downstream applications or not on V&L models.
2. This paper leverages bias amplification broadly examined in language models to understand to what extent the pertaining data cause the bias in a V&L model.
3. This paper further investigates the effect of continued pretraining V&L models on in-domain data where generated terms have been replaced by their gender-neutral counterparts.
4. This paper provides a clear demonstration of understanding bias in V&L models: (1) using gender-context distribution to define intrinsic bias; (2) leveraging BiasAmp metric to decouple the direction of bias amplification; (3) understanding extrinsic bias and fairness by evaluating model performance concerning the three demographic groups and comparing results in search of the more equitable system; (4) applying gender-neutral domain adaptation to reduce intrinsic and extrinsic biases.
5. This paper conducts comprehensive experiments on three models (LXMERT, ALBEF, and BLIP) under VOLTA framework among four evaluation tasks (MLM, VQA, NLVR, and image-text retrieval.) These experimental results strongly support the arguments and the detailed explanations help readers better understand.
6. Overall, this paper presents a comprehensive analysis of gender bias amplification and fairness of encoder-only and encoder-decoder V&L models, showing a promising direction for the community to leverage the evaluation methods from language-only to the vision-and-language area.



**Reasons To Reject:**

In my opinion, there are no significant reasons to reject this paper. The only thing I would worry about is the novelty. But it is reasonable to follow the wide-acknowledged methods in the community to explore the new area.

**Reproducibility:**

4: Could mostly reproduce the results, but there may be some variation because of sample variance or minor variations in their interpretation of the protocol or method.

**Reviewer Confidence:**

3: Pretty sure, but there's a chance I missed something. Although I have a good feel for this area in general, I did not carefully check the paper's details, e.g., the math, experimental design, or novelty.

---

> ### Author Rebuttal · Authors · 2023-08-28
>
> Thank you for your positive review and valuable feedback. We address your questions below.
>
> > [...] I am confused about how to summarize the results from Appendix E and Figure 4. Can you provide more illustrations or a clearer explanation?
>
> While it is true that “there is no evidence that the gender-neutral pertaining influenced positively (nor negatively) the extrinsic bias of the models”, it does not mean that the gender-neutral pretraining did not influence the extrinsic bias at all. In fact, as we can see in Figure 4 for GQA, there is a variation on the bias amplification measured on models pretrained on gender neutral data (M^N) compared to the original models (M), but this variance does not always go in the same direction: it is model and group dependant. In addition, the variance observed on M^N models also depends on the downstream task (see Appendix D). Therefore, we cannot establish a strong relationship between intrinsic and extrinsic biases (nor on the effect of gender-neutral pertaining) but conclude that they are independent.
>
> Regarding the results from Appendix E, we report variance of bias amplification (as recommended by [1]) in Table 8, where we can see that results are in line with what we explained earlier, i.e., there is no clear trend when comparing a model M with it’s gender-neutral pretraining counterpart, M^N. What’s more, the difference in the degree of bias amplification is minimal when taking into account the standard deviation. In addition, we also report fine-tuning variance in model performance across tasks (Figure 10). Here, we do observe a common trend: gender-neutral models reveal lower standard deviation across tasks. This finding indicates that it is beneficial to perform extra steps of pretraining on gender-neutral data to reduce variance in downstream performance. This observation aligns with the NLP literature showing that biases in a model are independent from model performance [2,3].
>
> We will clarify these points further when we revise the paper.
>
> References:
>
> [1] Angelina Wang and Olga Russakovsky. 2021. Directional bias amplification.
> [2] Aili Shen, Xudong Han, Trevor Cohn, Timothy Bald- 926 win, and Lea Frermann. 2022. Does representational fairness imply empirical fairness?
> [3] Laura Cabello, Anna Katrine Jørgensen, and Anders Søgaard. 2023. On the independence of association bias and empirical fairness in language models.

---

### Official Review · Reviewer_mmdc · 2023-08-12

**Soundness:** 4

**Excitement:**

3: Ambivalent: It has merits (e.g., it reports state-of-the-art results, the idea is nice), but there are key weaknesses (e.g., it describes incremental work), and it can significantly benefit from another round of revision. However, I won't object to accepting it if my co-reviewers champion it.

**Missing References:**

Several works in bias research have addressed the issue of intersectionality. This work only examines gender bias but the bias can exhibit in different ways when gender overlaps with other dimensions of identities. For instance, gender and race, gender and age, etc. The authors mentioned in the Limitation section that this framework can be extended to examine other dimensions of bias. It would make the paper stronger if the authors also address the limitation on not examining overlapping dimensions of identity. A few references on intersectionality in bias research:

Intersectionality:
Crenshaw, Kimberlé Williams. “Mapping the margins: intersectionality, identity politics, and violence against women of color.” Stanford Law Review 43 (1991): 1241-1299.

Intersectionality in bias research in NLP:
Field, A., Blodgett, S. L., Waseem, Z., & Tsvetkov, Y. (2021). A survey of race, racism, and anti-racism in NLP. arXiv preprint arXiv:2106.11410.

Tan, Yi Chern and Elisa Celis. “Assessing Social and Intersectional Biases in Contextualized Word Representations.” ArXiv abs/1911.01485 (2019): n. Pag.

Davidson, Thomas et al. “Racial Bias in Hate Speech and Abusive Language Detection Datasets.” ArXiv abs/1905.12516 (2019): n. Pag.

Lin, Inna Wanyin et al. “Gendered Mental Health Stigma in Masked Language Models.” Conference on Empirical Methods in Natural Language Processing (2022).


**Paper Topic And Main Contributions:**

The paper examined bias amplification in the pretraining and fine-tuning phase of V&L models and the possible connection between two training phases. The paper analyzed from intrinsic bias (bias encoded after the pretraining phase) and the extrinsic bias and task performance (bias encoded after the fine-tuning phase). The paper measures bias amplification on the intrinsic bias of the models and all three of bias amplification, task performance, and fairness on the extrinsic bias of fine-tuned models. The authors did not find causal connection between the intrinsic and extrinsic bias amplification and showed that extra steps of pretraining with gender neutral reduces group disparities.


**Questions For The Authors:**

Would it be possible for the authors to clarify how the intrinsic bias reflected in V&L models is different from the intrinsic biases in pretrained language models? Since the calculation of the intrinsic biases for V&L models is based on its language modeling capabilities, I wonder how much of it is due to the intrinsic bias from the language encoder and how much of it is from the cross-modality encoder. Is there a way to measure this? I believe this kind of analysis is informative and will also help ground V&L bias research in relation to existing language model bias research and vision models bias research.

**Reasons To Accept:**

The main strengths I find from the paper:

- The paper follows theoretically valid bias research framework and comprehensively examines bias from pretraining to downstream tasks.
- The paper asks and addresses interesting questions on top of examining the biases: are there correlations between the intrinsic and extrinsic bias amplification
- The paper also shows a viable solution to existing biases - adding extra training step with gender-neutral data and this comes with little costs of performance
- The experiments and well-designed and clearly communicated

**Reasons To Reject:**

I do not find significant weakness of the paper as reasons to reject. If the authors address the questions below that would make the paper stronger.

Minor weakness:
In the abstract the authors used the term "causal link" - this is slightly confusing. The results show that the intrinsic and extrinsic biases are independent (not correlated). "not causal" is a subset of "not correlated". I believe changing the word in the abstract makes the communication more accurate.

**Reproducibility:**

4: Could mostly reproduce the results, but there may be some variation because of sample variance or minor variations in their interpretation of the protocol or method.

**Reviewer Confidence:**

4: Quite sure. I tried to check the important points carefully. It's unlikely, though conceivable, that I missed something that should affect my ratings.

---

> ### Author Rebuttal · Authors · 2023-08-28
>
> Thank you for your positive review and valuable feedback. We address your questions below.
>
> > Minor weakness: In the abstract the authors used the term "causal link" - this is slightly confusing. The results show that the intrinsic and extrinsic biases are independent (not correlated). "not causal" is a subset of "not correlated". I believe changing the word in the abstract makes the communication more accurate.
>
> We agree with you and we will rephrase this sentence to convey the finding that intrinsic and extrinsic biases are independent.
>
> > Would it be possible for the authors to clarify how the intrinsic bias reflected in V&L models is different from the intrinsic biases in pretrained language models? [...]
>
> Thank you for raising this question, we will better clarify this methodology when describing the evaluation tasks (section 5.3). The intrinsic bias reflected by V&L models in our paper have been analyzed considering both domains, image and language. Following previous work [1,2,3,4,5] we also use the MLM task for evaluating intrinsic bias, and keep the image associated with each text (caption) as its input, in a setup that reflects the pretraining conditions (see L358–369 in the manuscript).
>
> Evaluating the linguistic skills of V&L models isn’t a trivial matter. We refer to ​​[7] for a detailed study on how to adapt pretrained V&L models to a text-only input. Early on in the project we performed an ablation study with LXMERT, where we zeroed-out the input image (we simply input a constant array of zeros) but kept the text component. We observed a higher degree of gender bias in the model compared to using both visual and text components.
>
> Therefore, we can say that the prejudicial biases found in V&L are not only attributed to one domain, but they are compound. For example, [5] studied how within- and cross-modality gender biases are expressed using a set of template-based data, and demonstrated that V&L models tend to reinforce a stereotype over faithfully describing the visual scene. In the field of image generation, [6] demonstrates the presence of stereotypes in image generation models and discuss the challenges of the compounding nature of language-vision biases.
>
> > Several works in bias research have addressed the issue of intersectionality. [...]
>
> Yes, thank you for pointing out the issue of intersectionality. We will acknowledge the fact that intersectionality was not studied in the limitations of the paper.
>
> References:
>
> [1] Jieyu Zhao, Tianlu Wang, Mark Yatskar, Vicente Ordonez, and Kai-Wei Chang. 2017. Men also like shopping: Reducing gender bias amplification using corpus-level constraints.
> [2] Lisa Anne Hendricks, Kaylee Burns, Kate Saenko, Trevor Darrell, and Anna Rohrbach. 2018. Women also snowboard: Overcoming bias in captioning models.
> [3] Ruixiang Tang, Mengnan Du, Yuening Li, Zirui Liu, Na Zou, and Xia Hu. 2021. Mitigating gender bias in captioning systems.
> [4] Tejas Srinivasan and Yonatan Bisk. 2021. Worst of both worlds: Biases compound in pre-trained vision-and- language models.
> [5] Tianlu Wang, Jieyu Zhao, Mark Yatskar, Kai-Wei Chang, and Vicente Ordonez. 2019. Balanced datasets are not enough: Estimating and mitigating gender bias in deep image representations.
> [6] Federico Bianchi, Pratyusha Kalluri, Esin Durmus, Faisal Ladhak, Myra Cheng, Debora Nozza, Tatsunori Hashimoto, Dan Jurafsky, James Zou, and Aylin Caliskan. 2023. Easily Accessible Text-to-Image Generation Amplifies Demographic Stereotypes at Large Scale.
> [7] Lovisa Hagström and Richard Johansson. 2022. How to Adapt Pre-trained Vision-and-Language Models to a Text-only Input?.

---

### Meta-Review · Area_Chair_LWCq · 2023-09-18

**Recommendation:** 5

**Metareview:**

This paper analyzes different types (intrinsic and extrinsic) of gender bias amplification correlating to different stages (pre-training and fine-tuning) of training vision and language models. The paper also proposes a method for addressing bias amplification through an additional pre-training step. Reviewers suggest including a limitation in the work that it focuses on a single dimension of social bias (gender), and doesn't consider how different dimensions of identity may intersect with one another in nuanced ways (e.g., race and gender). Other suggestions that I encourage the authors to make include clarifying some of the notation and expanding some of the discussion on related work.

---

### Decision · Program_Chairs · 2023-10-07

**Decision:**

Accept-Main

**Comment:**

This paper analyzes different types (intrinsic and extrinsic) of gender bias amplification correlating to different stages (pre-training and fine-tuning) of training vision and language models. The paper also proposes a method for addressing bias amplification through an additional pre-training step. Reviewers suggest including a limitation in the work that it focuses on a single dimension of social bias (gender), and doesn't consider how different dimensions of identity may intersect with one another in nuanced ways (e.g., race and gender). Other suggestions that I encourage the authors to make include clarifying some of the notation and expanding some of the discussion on related work.